# Conditional Networks for Few-Shot Semantic Segmentation

**Kate Rakelly   Evan Shelhamer   Trevor Darrell   Alexei Efros   Sergey Levine**
UC Berkeley
{rakelly,shelhamer,efros,slevine,trevor}@eecs.berkeley.edu

## Abstract

Few-shot learning methods aim for good performance in the low-data regime. Structured output tasks such as segmentation present difficulties for few-shot learning because of their high dimensionality and the statistical dependencies among outputs. To tackle this problem, we propose the co-FCN, a conditional network learned by end-to-end optimization to perform fast, accurate few-shot segmentation. The network conditions on an annotated support set of images via feature fusion to do inference on an unannotated query image. Once learned, our conditioning approach requires no further optimization for new data. Annotations are instead conditioned on in a single forward pass, making our method suitable for interactive use. We evaluate our co-FCN with dense and sparse annotations, and it achieves competitive accuracy even when given only one positive pixel and one negative pixel, reducing the annotation burden for segmenting new concepts.

## 1 Introduction

Convolutional networks are driving progress in the visual recognition of what and where, enabled partly by large labeled datasets that are expensive and time consuming to collect. Few-shot learning holds the promise of data efficiency; in the extreme case, one-shot learning requires only a single annotation of a new concept. Some current methods rely on meta-learning, or learning to learn, in order to quickly adapt to new domains or tasks. While these methods are promising, the focus has been on classification, and little has been done for structured output tasks. Current methods largely cannot be applied out-of-the-box to the structured output setting due to the high dimensionality of the output space, as well as the statistical dependencies among outputs that result from the spatial correlation of pixels in the input.

Semantic segmentation is a challenging core task for visual recognition. Networks optimized end-to-end have achieved state-of-the-art performance, but rely on large pixel-wise labeled datasets that are particularly onerous to collect, rendering any relief in the annotation burden practically significant. We therefore address the problem of few-shot semantic segmentation, as first proposed by Shaban et al. (2017). In our co-FCN network, we augment the FCN (Shelhamer et al., 2016) architecture with a conditioning branch to incorporate the few-shot annotations. No gradients flow at test time; given a new few-shot task, solving it is a single forward pass in the network. During training, we simulate few-shot tasks by sampling them from a densely labeled semantic segmentation dataset.

Our work is related to one-shot and interactive approaches to segmentation. Shaban et al. (2017) are the first to address few-shot semantic segmentation. They assume dense pixel-wise few-shot annotations. Our method achieves nearly the same accuracy with only one positive and one negative pixel. Caelles et al. (2017) show the effectiveness of fine-tuning for video object segmentation, but requiring optimization for every input at test time is too costly in computation and annotation. Xu et al. (2016) meta-learn state-of-the-art interactive object segmentation, but are restricted to propagating annotations within a single image, and cannot propagate across images. Our contributions cover handling sparse pixel-wise annotations, conditioning features vs. parameters, and evaluating stronger segmentation and meta-learning baselines.

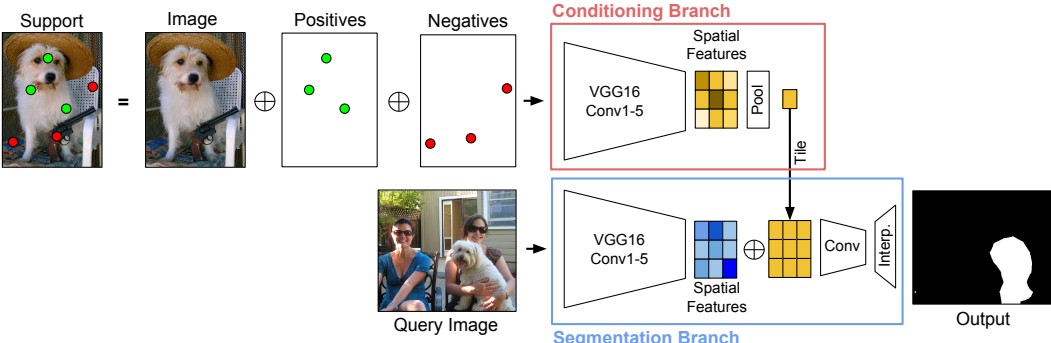

Figure 1: co-FCN performs few-shot segmentation in a single forward pass of the network. The conditioning branch (top) takes the support image and (dense or sparse) annotations concatenated channel-wise and encodes them as features (illustrated here) or parameters. The segmentation branch (bottom) conditions on this encoding to densely segment the query. For training (not shown), few-shot tasks are synthesized from a densely labeled dataset. The loss compares the predicted segmentation of the query image with the target, which is defined jointly by the support and the ground truth semantic segmentation of the query. The two branches are learned jointly and end-to-end.

## 2   CONDITIONAL ARCHITECTURE AND FEW-SHOT OPTIMIZATION

Few-shot segmentation aims to learn to segment new concepts, given only a few annotated examples of those concepts. A $k$-shot task consists of the labeled *support set* of $k$ image-annotation pairs, and the unlabeled *query*. The segmentation task adds a new dimension to the classic few-shot setup, as the support set annotations may be spatially dense or sparse. We explore both and show that the co-FCN is more robust to sparse annotations than other methods.

We adapt the fully convolutional network (FCN) approach for image-to-image tasks (Shelhamer et al., 2016) to the few-shot setting. In practice, FCNs deliver accurate and coherent pixel-wise predictions, but they require too many parameters for fine-tuning to succeed as a few-shot learning method. Instead, we align the few-shot training and testing paradigms to produce a model that requires no optimization at test time.

The co-FCN is shown in schematic form in Figure 1. The segmentation branch, given conditional features or parameters, is a standard FCN that segments the query. The conditioning branch takes the support set as input and makes a globally pooled prediction that informs the segmentation of the query. This prediction may be features which are then fused with query features for further processing. Alternatively, the conditioning branch may predict the parameters of the linear classification layer at the end of the query branch. We explore both choices in our experiments. Each branch is fully convolutional to decompose the problem and prevent overfitting.

For training, we synthesize few-shot segmentation tasks from a densely annotated semantic segmentation dataset. First, we sample a concept (e.g., "dog" then sample a support set of images that contain the concept. The annotations are ground truth semantic segmentation labels binarized to encode only the sampled concept. In our experiments with a sparsely annotated support set, the annotations are also sampled spatially. The cross-entropy loss is computed between the prediction for the query and its binarized label. The network is directly optimized for the few-shot task; conditioning and segmentation branches are jointly optimized end-to-end. We fine-tune from ILSVRC (2010-2012) pre-training to improve stability and data efficiency. To handle k-shot learning from a support set of size $k$, we simply average the conditional features computed independently for each member of the support. This averaged feature is then fused with the query branch to produce the $k$-shot prediction.

## 3   FEW-SHOT SEGMENTATION RESULTS

We show results on the PASCAL VOC dataset (Everingham et al.) and its further annotations from SBDD (Hariharan et al., 2011), as is standard for semantic segmentation, but cast into few-shot

binary segmentation tasks. See Table 1 for intersection-over-union scores. We follow the few-shot protocol of Shaban et al. (2017) by defining four splits of held-out classes, and constructing our training set from all images containing non-held-out classes. We remap any occurrences of held-out class annotations to background during training. For evaluation, we sample a support for every query that contains a held-out class in the standard non-intersecting validation set. To gauge the effect of conditioning, we score a foreground-background baseline that is independent of the support. All of our experiments measure generalization accuracy on the held-out classes.

| Method | Dense | | Sparse | |
|---|---|---|---|---|
| | 1-Shot | 5-Shot | 1-Shot | 5-Shot |
| FG-BG | 55.0 | - | - | - |
| Fine-tuning | 55.1 | 55.6 | | |
| Shaban et al. (2017) | 61.3 | 61.5 | 52.5 | 52.9 |
| co-FCN (feat.) | 60.1 | 60.2 | 60.1 | 58.4 |
| co-FCN (param.) | 58.3 | 58.3 | | |
| Oracle | 80.8 | - | - | - |

Table 1: Few-shot segmentation evaluation on PASCAL VOC with the intersection-over-union (IU) metric over binary tasks. We follow the experimental protocol of Shaban et al. (2017) and report the IU achieved by each method averaged across four class-wise splits. We focus evaluation on both dense and sparse support annotations with full masks and a single point per positive/negative respectively. Our co-FCN results are shaded: we achieve nearly state-of-the-art accuracy with just two labeled pixels. Note that FG-BG is a strong baseline and rivals fine-tuning. Oracle is trained for semantic segmentation on all of the classes in the dataset and scored on the same binary tasks by picking the output mask corresponding to the support class.

**Baselines and Prior Methods** We compare to fine-tuning, current few-shot methods for general use, the prior work on few-shot segmentation, and foreground-background segmentation.

- Fine-tuning: Fine-tuning is a standard approach to transfer learning. We follow Caelles et al. (2017) by fine-tuning a pre-trained foreground-background segmentor on the support.

- MAML Finn et al. (2017): In preliminary experiments fine-tuning from MAML did not give any improvement over fine-tuning from ILSVRC pre-training.

- Global Parameter Prediction: Shaban et al. (2017) meta-learn to regress the score layer parameters of a segmentor. Their predictor is global instead of local and fully convolutional. They make use of higher dimensional parameters and so rely on weight hashing to reduce overfitting. For comparability of results, we evaluate their reference models with the same standard data and binary IU metric as our method.

- Foreground-Background: To check that co-FCN makes use of the support, we compare against a foreground-background segmentor that is independent of the support. We train for binary segmentation on the union of training classes for each split.

## 4 DISCUSSION

Fully convolutional conditioning is state-of-the-art for learning how to guide fast and accurate few-shot segmentation from sparse annotations. In addition to being more accurate than fine-tuning, our conditional model is also much faster because it requires no optimization at test time. The ability to cope with sparse annotations makes the co-FCN particularly well-suited for interactive use, such as propagating labels to segment a personal photo collection. Our method could also be applied to automatically annotate unlabeled data. Further experiments are required to assess the ability of the model to transfer to unseen datasets.

Future work could investigate the generality of few-shot conditioning with respect to the type of segmentation task. For instance, video object segmentation requires instance-wise propagation of annotations instead of class-wise propagation as done here. Extending to different strengths of support supervision, such as scribbles or bounding boxes, is also an exciting direction.

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
