# OpenReview forum: "Conditional Networks for Few-Shot Semantic Segmentation"
_ICLR.cc/2018/Workshop — Accept_

### Official Review · AnonReviewer1 · 2018-03-05
**Good paper, accept**

**Rating:** 7
**Confidence:** 4

**Review:**

This paper proposed a conditional network for k-shot semantic segmentation. The main idea is in line with the recent few-shot learning methods in the sense that a meta model is used to regress the parameters or latent features of the learner model — in the context of this paper, a FCN. Results show that the proposed method outperforms existing ones under a convincing experiment setup. It is a nice extension of the few-shot learning regime from recognition problems to highly structured semantic segmentation. The authors are recommended to release an extended version to describe the details of the implementation and experiments.

---

### Official Review · AnonReviewer3 · 2018-03-08
**Good work**

**Rating:** 7
**Confidence:** 4

**Review:**

Interesting contribution and outperformed a previous result. Few-shot segmentation is an interesting new domain to work on.

---

### Author Response · Authors · 2018-05-09
**Revision with Results on Sparse Support Annotations and Standardized Evaluation**

We extend our method to sparse support annotations which in the extreme case are limited to only one positive pixel and one negative pixel. The accuracy of our co-FCN in this sparse regime is comparable to the dense regime while drastically reducing the annotation burden, and furthermore it is more robust to sparsity than the prior method for few-shot semantic segmentation.

We standardize the data and metric in the evaluation for all methods. At the time of submission to the workshop, the choice of data and metric used by Shaban et al. 2017 were not perfectly clear, but since then the public release of a reference implementation and correspondence with the authors allowed us to reproduce their method exactly. With this reproduction we were able to evaluate their method in the same experimental framework as the rest of our comparisons:

- the data is the "valid" split of PASCAL VOC 2012, which is the set of val images that do not intersect with SBDD train, as detailed at https://github.com/shelhamer/fcn.berkeleyvision.org/tree/master/data/pascal
- the metric is the intersection-over-union (IU) averaged over the positive and negative classes, analogous to the mean IU for semantic segmentation, which is averaged over all classes including background
- we select every input in the "valid" split as a query and randomly select a support for it from the pool of inputs in the "valid" split containing the same class and fix this assignment of query-support pairs across all methods

The results in Table 1 are updated accordingly: in the dense regime our method is competitive to the prior method and in the sparse regime our method is state-of-the-art.

---

### Author Response · Authors · 2018-06-20
**Full Edition on ArXiv**

The full edition of this work is on arxiv at https://arxiv.org/abs/1806.07373 with code and models at https://github.com/shelhamer/revolver

We propose guided networks, which extract a latent task representation from any amount of supervision, and optimize our novel architecture end-to-end for fast, accurate few-shot segmentation for semantic segmentation, video object segmentation, and real-time interactive segmentation of images and videos.

---

### Decision · Program_Chairs · 2018-03-20
**ICLR 2018 Workshop Acceptance Decision**

**Decision:**

Accept

**Comment:**

Congratulations, your paper was accepted to the ICLR workshop.